# DINO: DETR with Improved DeNoising Anchor Boxes for End-to-End Object Detection

## Abstract

We present DINO (**D**ETR with **I**mproved de**N**oising anch**O**r boxes), a strong end-to-end object detector. DINO improves over previous DETR-like models in performance and efficiency by using a contrastive way for denoising training, a look forward twice scheme for box prediction, and a mixed query selection method for anchor initialization. DINO achieves $49.4$AP in 12 epochs and $51.3$AP in 24 epochs on COCO with a ResNet-50 backbone and multi-scale features, yielding a significant improvement of **+6.0AP** and **+2.7AP**, respectively, compared to DN-DETR, the previous best DETR-like model. DINO scales well in both model size and data size. Without bells and whistles, after pre-training on the Objects365 dataset with a SwinL backbone, DINO obtains the best results on both COCO `val2017` (**63.2AP**) and `test-dev` (**63.3AP**) with model size under 1 billion parameters. Compared to other models on the leaderboard, DINO achieves better results with smaller model size and pre-training data size. The code will be available.

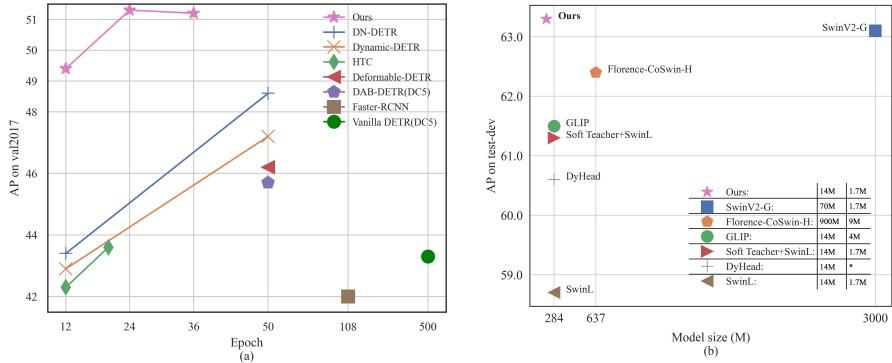

Figure 1: AP on COCO compared with other detection models. (a) Comparison to models with a ResNet-50 backbone w.r.t. training epochs. Models marked with DC5 use a dilated larger resolution feature map. Other models use multi-scale features. (b) Comparison to SOTA models w.r.t. pre-training data size and model size. SOTA models are from the COCO `test-dev` leaderboard. In the legend we list the backbone pre-training data size (first number) and detection pre-training data size (second number). $*$ means the data size is not disclosed.

## 1 Introduction

Object detection is a fundamental task in computer vision. Remarkable progress has been accomplished by classical convolution-based object detection algorithms (Ren et al., 2017; Tian et al., 2019; Lin et al., 2020; Bochkovskiy et al., 2020; Ge et al., 2021). Despite that such algorithms normally include hand-designed components like anchor generation and non-maximum suppression (NMS), they yield the best detection models such as DyHead (Dai et al., 2021a), Swin (Liu et al., 2021b) and SwinV2 (Liu et al., 2021a) with HTC++ (Chen et al., 2019a), as evidenced on the COCO test-dev leaderboard (pap).

In contrast to classical detection algorithms, DETR (Carion et al., 2020) is a novel Transformer-based detection algorithm. It eliminates the need of hand-designed components and achieves com-

parable performance with optimized classical detectors like Faster RCNN (Ren et al., 2017). Different from previous detectors, DETR models object detection as a set prediction task and assigns labels by bipartite graph matching. It leverages learnable queries to probe the existence of objects and combine features from an image feature map like soft ROI pooling (Liu et al., 2022).

Despite its promising performance, it converges slow and the meaning of queries is unclear. To address such problems, many methods have been proposed, such as introducing deformable attention (Zhu et al., 2021), decoupling positional and content information (Meng et al., 2021), providing spatial priors (Gao et al., 2021; Yao et al., 2021; Wang et al., 2021), etc. Recently, DAB-DETR (Liu et al., 2022) proposes to formulate DETR queries as dynamic anchor boxes (DAB), which bridges the gap between classical anchor-based detectors and DETR-like ones. DN-DETR (Li et al., 2022) further accelerate convergence by introducing a denoising (DN) technique. These improvements promote the development of DETR-like models, while it remains not on the list of first-choice detectors in the field.

The best detection models nowadays are based on improved classical detectors like DyHead (Dai et al., 2021b) and HTC (Chen et al., 2019a). For example, the best result presented in SwinV2 (Liu et al., 2021a) was trained with the HTC++ (Chen et al., 2019a; Liu et al., 2021b) framework. Two main reasons contribute to the phenomenon: 1) *Previous DETR-like models are inferior* to the improved classical detectors. Most classical detectors have been well studied and highly optimized, leading to a better performance compared with the newly developed DETR-like models. 2) *The performance of DETR-like model has not been tested* on large backbone with large-scale pre-training data. We aim to address both concerns in this paper.

Specifically, by improving the denoising training, query initialization, and box prediction, we design a new DETR-like model based on DN-DETR, DAB-DETR, and Deformable DETR. We name our model as **DINO** (**D**ETR with **I**mproved de**N**oising anch**O**r box). As shown in Fig. 1, the comparison on COCO shows the superior performance of DINO. In particular, DINO demonstrates a strong performance, setting a new record of 63.3 AP for models with less than 1 billion parameters on the COCO test-dev leaderboard (pap).

As a DETR-like model, DINO contains a backbone, a multi-layer Transformer encoder, a multi-layer Transformer decoder, and multiple prediction heads. Following DAB-DETR, we formulate queries in decoder as dynamic anchor boxes and refine them step-by-step across decoder layers. Following DN-DETR, we add ground truth labels and boxes with noises into the Transformer decoder layers to help stabilize bipartite matching during training. We also adopt deformable attention (Zhu et al., 2021) for its computational efficiency. Moreover, we propose three new methods as follows. First, to reduce duplicate predictions, we propose a *contrastive denoising training* by adding both positive and negative samples of the same ground truth at the same time. After adding two different noises to the same ground truth box, we mark the box with a smaller noise as positive and the other as negative. The contrastive denoising training helps the model to predict more precise boxes and avoid duplicate outputs of the same target. Second, to overcome the shortsightedness of refining boxes in each decoder layer, which is a greedy way proposed in Deformable DETR, while keeping the advantages of fast convergence, we propose a new *look forward twice* scheme to correct the updated parameters with gradients from later layers. Third, the dynamic anchor box formulation of queries links DETR-like models with classical two-stage models. Hence we propose a *mixed query selection* method, which helps better initialize the queries. We select initial anchor boxes as positional queries from the output of the encoder, similar to (Zhu et al., 2021; Yao et al., 2021). However, we leave the content queries learnable queries aligned with CDN part where queries are also learnable queries which encourages the first decoder layer to focus on the spatial prior.

We validate the effectiveness of DINO with extensive experiments on the COCO (Lin et al., 2014) detection benchmarks. As shown in Fig. 1, DINO achieves 49.4AP in 12 epochs and 51.3AP in 24 epochs with ResNet-50 multi-scale features, yielding a significant improvement of **+6.0**AP and **+2.7**AP, respectively, compared to the previous best DETR-like model DN-DETR. In addition, DINO scales well in both model size and data size. After pre-training on the Objects365 (Shao et al., 2019) data set with a SwinL (Liu et al., 2021b) backbone, DINO achieves impressive results on both COCO `val2017` (**63.2**AP) and `test-dev` (**63.3**AP) benchmarks, as shown in Table 4. Our DINO reduces the model size to **1/15** compared to SwinV2-G (Liu et al., 2021a). Moreover, DINO outperforms Florence (Yuan et al., 2021) with only **1/60** backbone pre-training dataset and **1/5** detection pre-training dataset.

To summarize, our contributions are three-fold. **1)** We design a new end-to-end DETR-like object detector with several novel techniques, including contrastive denoising training, look forward twice, and mixed query selection for different parts of the DINO model. **2)** We conduct intensive ablation studies to validate the effectiveness of different design choices in DINO. As a result, DINO achieves 49.4AP in 12 epochs and 51.3AP in 24 epochs with ResNet-50 and multi-scale features, significantly outperforming the previous best DETR-like model DN-DETR. **3)** We show that, without bells and whistles, DINO can achieve the best performance on public benchmarks with model size under 1 billion parameters. After pre-training on the Objects365 (Shao et al., 2019) dataset with a SwinL (Liu et al., 2021b) backbone, DINO achieves **63.2**AP on COCO `val2017` and **63.3**AP on COCO `test-dev` benchmarks.

## 2    RELATED WORK

**Classical Object Detectors:**  Early convolution-based object detectors are either two-stage or one-stage models, based on hand-crafted anchors or reference points. Two-stage models (Ren et al., 2015; He et al., 2017) usually use an region proposal network (RPN) (Ren et al., 2015) to propose potential boxes, which are then refined in the second stage. One-stage models (Redmon & Farhadi, 2017; 2018) directly output offsets relative to predefined anchors. Recently, some convolution-based models such as HTC++ (Chen et al., 2019a) and Dyhead (Dai et al., 2021a) have achieved top performance on the COCO 2017 (Lin et al., 2014). The performance of convolution-based models, however, rely on the way they generate anchors and need hand-designed components like NMS.

**DETR and Its Variants:**  Carion *et al.* (Carion et al., 2020) proposed a Transformer-based end-to-end object detector named DETR (DEtection TRansformer) without using hand-designed components like anchor design and NMS. Many follow-up papers have attempted to address the slow training convergence issue of DETR introduced by decoder cross-attention. For instance, Dai *et al.* (Dai et al., 2021a) proposed a dynamic decoder to focus on important regions from multiple feature levels. Another line of works is towards a deeper understanding of decoder queries in DETR. Many papers associate queries with spatial position from different perspectives. Deformable DETR (Zhu et al., 2021) predicts 2D anchor points and designs a deformable attention module that only attends to certain sampling points around a reference point. DAB-DETR (Liu et al., 2022) further extends 2D anchor points to 4D anchor box coordinates to represent queries and dynamically update boxes in each decoder layer. Recently, DN-DETR (Li et al., 2022) introduces a denoising training method to speed up DETR training. It feeds noise-added ground-truth labels and boxes into the decoder and trains the model to reconstruct the original ones. Our work is based on DAB-DETR and DN-DETR, and also adopts deformable attention for its computational efficiency.

**Large-scale Pre-training for Object Detection:**  The best performing detectors nowadays are mostly achieved with large backbones pre-trained on large-scale data. For example, Swin V2 (Liu et al., 2021a) extends its backbone size to 3.0 billion parameters and pre-trains its models with 70M privately collected images. Florence (Yuan et al., 2021) first pre-trains its backbone with 900M privately curated image-text pairs and then pre-trains its detector with 9M images with annotated or pseudo boxes. In contrast, DINO achieves better results with a publicly available SwinL (Liu et al., 2021b) backbone and a public dataset Objects365 (Shao et al., 2019) (1.7M annotated images) only.

## 3    DINO: DETR WITH IMPROVED DENOISING ANCHOR BOXES

### 3.1    PRELIMINARIES

As studied in Conditional DETR (Meng et al., 2021) and DAB-DETR (Liu et al., 2022), queries in DETR (Carion et al., 2020) are formed by two parts: a positional part and a content part, which are referred to as positional queries and content queries in this paper. DAB-DETR explicitly formulates each positional query in DETR as a 4D anchor box $(x, y, w, h)$, where $x$ and $y$ are the center coordinates of the box and $w$ and $h$ correspond to its width and height. Such an explicit anchor box formulation makes it easy to dynamically refine anchor boxes layer by layer in the decoder.

DN-DETR (Li et al., 2022) introduces a denoising (DN) training method to accelerate the training convergence of DETR-like models. It shows that the slow convergence problem in DETR is caused by the instability of bipartite matching. To mitigate this problem, DN-DETR proposes to

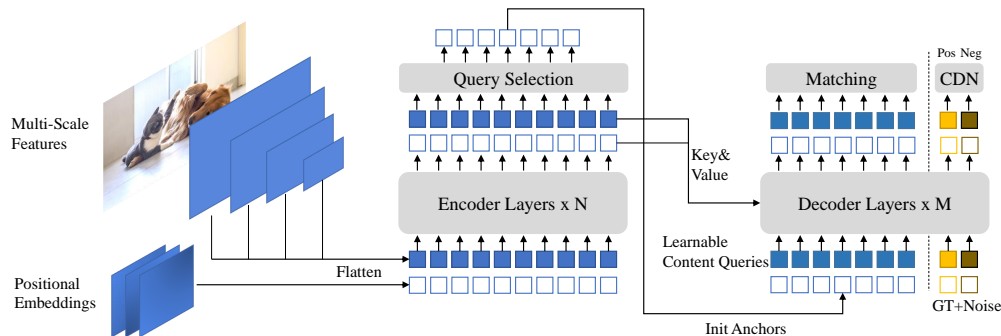

Figure 2: The framework of our proposed DINO model. Our improvements are mainly in the Transformer encoder and decoder. The top-K encoder features in the last layer are selected to initialize the positional queries for the Transformer decoder. Our decoder also contains a Contrastive DeNoising (CDN) part with both positive and negative examples.

additionally feed noised ground-truth (GT) labels and boxes into the Transformer decoder and train the model to reconstruct the ground-truth ones. The noise $(\Delta x, \Delta y, \Delta w, \Delta h)$ is constrained by $|\Delta x| < \frac{\lambda w}{2}$, $|\Delta y| < \frac{\lambda h}{2}$, $|\Delta w| < \lambda w$, and $|\Delta y| < \lambda h$, where $(x, y, w, h)$ denotes a GT box and $\lambda$[1] is a hyper-parameter to control the scale of noise. Since DN-DETR view decoder queries as anchors, a noised GT box can be viewed as a special anchor with a GT box nearby as $\lambda$ is usually small. In addition to the orginal DETR queries, DN-DETR adds a DN part which feeds noised GT labels and boxes into the decoder to provide an auxiliary DN loss. The DN loss effectively stabilizes and speeds up the DETR training and can be plugged into any DETR-like models.

Deformable DETR (Zhu et al., 2021) is another early work to speed up the convergence of DETR. To compute deformable attention, it introduces the concept of reference point so that deformable attention can attend to a small set of key sampling points around a reference. The reference point concept makes it possible to develop several techniques to further improve the DETR performance. The first technique is query selection (or "two stage"), which selects features and reference boxes from the encoder as inputs to the decoder directly. The second technique is iterative bounding box refinement with a careful gradient detachment design between two decoder layers. We call this gradient detachment technique "look forward once" in our paper.

Following DAB-DETR and DN-DETR, DINO formulates the positional queries as dynamic anchor boxes and is trained with an extra DN loss. DINO additionally introduces three methods, which will be described in Sec. 3.3, Sec. 3.4, and Sec. 3.5, respectively.

## 3.2 MODEL OVERVIEW

As a DETR-like model, DINO is an end-to-end architecture which contains a backbone, a multi-layer Transformer (Vaswani et al., 2017) encoder, a multi-layer Transformer decoder, and multiple prediction heads. The overall pipeline is shown in Fig. 2. Given an image, we extract multi-scale features with a backbone, and then feed them into the Transformer encoder with corresponding positional embeddings. After feature enhancement with the encoder layers, we propose a new mixed query selection strategy to initialize anchors as positional queries for the decoder. Note that this strategy does not initialize content queries but leaves them learnable. More details of mixed query selection are available in Sec. 3.5. With the initialized anchors and the learnable content queries, we use the deformable attention (Zhu et al., 2021) to combine the features of the encoder outputs and update the queries layer-by-layer. The final outputs are formed with refined anchor boxes and classification results predicted by refined content features. As in DN-DETR, we have an extra DN branch to perform denoising training. Beyond the standard DN method, we propose a new contrastive denoising training approach by taking into account hard negative samples, which will be presented in Sec. 3.3. To overcome the shortsightedness of the greedy way for box refinement in previous works, a novel look forward twice method is proposed to pass gradients between adjacent layers, which will be described in Sec. 3.4.

---

[1]The DN-DETR paper (Li et al., 2022) uses $\lambda_1$ and $\lambda_2$ to denote noise scales of center shifting and box scaling, but sets $\lambda_1 = \lambda_2$. In this paper, we use $\lambda$ in place of $\lambda_1$ and $\lambda_2$ for simplicity.

## 3.3 CONTRASTIVE DENOISING TRAINING

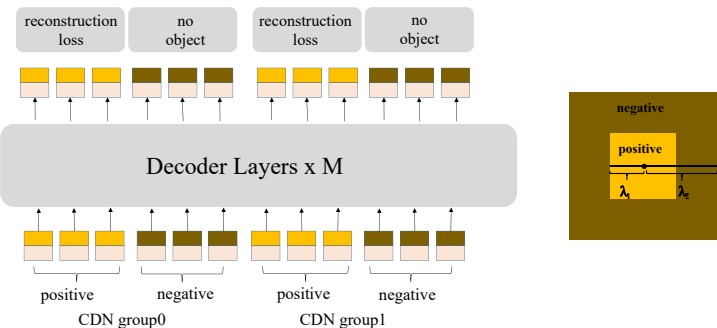

Figure 3: The structure of CDN group and a demonstration of positive and negative examples. Although both positive and negative examples are 4D anchors that can be represented as points in 4D space, we illustrate them as points in 2D space on concentric squares for simplicity. Assuming the square center is a GT box, points inside the inner square are regarded as a positive example and points between the inner square and the outer square are viewed as negative examples.

DN-DETR is effective in stabilizing training and accelerating convergence. With the help of De-Noising (DN) queries, it learns to make predictions based on noised Ground-Truth (GT) boxes, which leads to fast convergence. However, each DN query in DN-DETR is matched with a GT box and lacks the ability to predict background for "no object". Since predicting background is also important for DETR-like model to reduce duplicate predictions, we propose a Contrastive DeNoising (CDN) approach to *rejecting* hard negative examples. To maximize the utilization of denoising queries, we also propose to use adaptive number of denoising groups.

**Implementation:** DN-DETR has a hyper-parameter $\lambda$ to control the noise scale. The generated noises are no larger than $\lambda$ as DN-DETR wants the model to reconstruct the ground truth (GT) from moderately noised queries. In our method, we have two hyper-parameters $\lambda_1$ and $\lambda_2$, where $\lambda_1 < \lambda_2$. As shown in the concentric squares in Fig. 3, we generate two types of CDN queries: positive queries and negative queries. Positive queries within the inner square have a noise scale smaller than $\lambda_1$ and are expected to reconstruct their corresponding ground truth boxes. Negative queries between the inner and outer squares have a noise scale larger than $\lambda_1$ and smaller than $\lambda_2$. They are expected to predict "no object". We usually adopt a small $\lambda_2$ because hard negative samples closer to GT boxes can better help the model suppress duplicate predictions. As shown in Fig. 3, each CDN group has a set of positive queries and negative queries. If an image has $n$ GT boxes, a CDN group will have $2 \times n$ queries with each GT box generating both a positive and a negative queries. Similar to DN-DETR, we also use multiple CDN groups to improve the effectiveness of our method. The reconstruction losses are $l_1$ and GIOU losses for box regression and focal loss (Lin et al., 2020) for classification. The loss to classify negative samples as background is also focal loss. Furthermore, to better utilize DN queries. We improve DN-DETR's design of using a fixed number of denoising groups with an adaptive number of denoising groups. For each image, we fix the total number of denoising queries as $N$. For an image with $n$ objects, the number of CDN groups is $\frac{N}{2n}$.

**Analysis:** The reason why CDN works is because it explicitly introduces hard negative examples that are very similar to positive example. Such negative examples encourage the model to learn subtle differences between positive and negative boxes for more precise box predictions. The ability to distinguish positive and negative example also enables the model to further reduce duplicate predictions on the basis of DETR. DETR eliminates the need of using NMS to suppress duplicate boxes. Instead, it relies on bipartite matching to pick up only one query for each GT box and suppress other queries by pushing them away or lowering their confidence. However, the suppressed queries are normally not hard negative. As a result, DETR cannot completely avoid duplicate boxes, especially for low confidence boxes. CDN addresses this issue by introducing explicitly designed negative queries, which further enhance the effect of bipartite matching on avoiding duplicate boxes. For example, on the COCO dataset, we compare CDN with its counterpart DN, both using 300 predictions. The numbers of duplicate predictions for each method are shown in Table 3.3. For all the thresholds from 0 to 0.3, CDN constantly predicts fewer duplicate boxes than DN.

| | Threshold | 0.00 | 0.05 | 0.10 | 0.15 | 0.20 | 0.25 | 0.30 |
|---|---|---|---|---|---|---|---|---|
| DN | total | 292.65 | 158.51 | 59.91 | 24.16 | 10.26 | 3.94 | 0.52 |
| | duplicate | 67.91 | 31.53 | 9.63 | 3.53 | 1.30 | 0.53 | 0.26 |
| CDN | total | 292.65 | 164.62 | 60.21 | 23.45 | 9.90 | 3.83 | 0.62 |
| | duplicate | 53.72 | 25.87 | 7.35 | 2.55 | 0.97 | 0.40 | 0.20 |

Table 1: For a fair comparison, we only change CDN to DN and keep other hyper-parameters unchanged. For each model, we choose the top 300 predictions and filter them according to confidence scores with 7 thresholds from 0 to 0.3. For each threshold $t_i$, "total" and "duplicate" denote the numbers of total and duplicate predictions with scores greater than $t_i$, respectively. We view predictions with IoU $> 0.8$ as duplicate predictions.

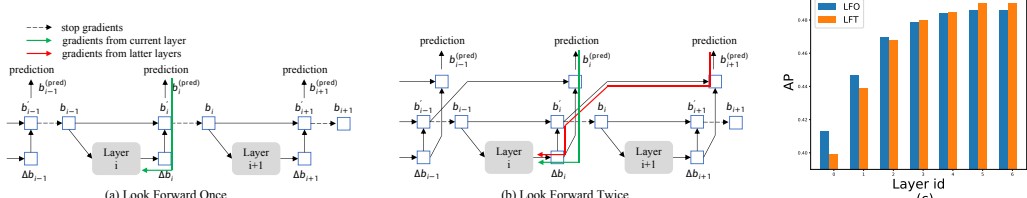

Figure 4: (a)(b) Comparison of box update in Deformable DETR and our method. (c) APs of look forward once and look forward twice in each decoder layer. "LFO" and "LFT" denote look forward once and look forward twice, respectively.

## 3.4 LOOK FORWARD TWICE

We propose a new approach to improving box prediction in this section. The iterative box refinement in Deformable DETR blocks gradient back propagation to stabilize training. We name the method look forward once since the parameters of the $i$-th decoder layer $L_i$ are updated based on the auxiliary loss of boxes $b_i^{(pred)}$ only, as shown in Fig 4 (a), where $b_i^{(pred)}$ denotes the predicted boxes in $L_i$. Such a parameter update approach is a greedy method, in which each decoder layer approximates ground truth boxes individually while trying not to influence its previous layers by blocking gradient. Such a method stabilizes training and helps convergence in early training stages. However, it may lead to a sub optimal result. On the other hand, allowing gradient to propagate from all latter layers will make the model hard to converge. To address this issue, we propose to only allow $L_{i-1}$ to be influenced by gradients from itself and $L_i$ as shown in 4 (b). Since parameters in $L_{i-1}$ are optimized to approximate ground truth boxes in both $L_{i-1}$ and $L_i$, we name our method as look forward twice, which is more comprehensive compared with the look forward once method.
**Implementation:** We compare the implementations of look forward once (LFO) and look forward twice (LFT) as follows. Since LFO and LFT share the same process from $b'_{i-1}$ to $b'_i$, we first show this process. Denote $b'_{i-1}$ and $b_{i-1}$ as the boxes before and after stopping gradient. We have

$$b_{i-1} = \text{sg}\left[b'_{i-1}\right],\tag{1}$$

where sg[·] denotes stopping gradient. $b_{i-1}$ is used as the input anchor box in $L_i$ to obtain $\Delta b_i$ as follows.

$$\Delta b_i = L_i(b_{i-1}; \theta_i),\tag{2}$$

where $L_i$ denotes the $i$-th Decoder layer with $\theta_i$ as its parameters. We ignore other inputs to $L_i$ for simplicity. $b'_i$ is obtained as follows.

$$b'_i = \sigma\left(\sigma^{-1}(b_{i-1}) + \Delta b_i\right).\tag{3}$$

where $\sigma(\cdot)$ and $\sigma^{-1}$ denote the sigmoid and inverse sigmoid functions. Note that such a box update approach is to guarantee that the updated boxes have normalized $x, y, w, h$ values between 0 and 1. Equation 3 is marked with green line in Fig. 4(a) where gradients are propagated from $b_i^{(pred)}$ to $\theta_i$ through $\Delta b_i$. In LFO, the prediction $b_i^{(pred)}$ is equal to $b'_i$. While in LFT, we update box predictions $b_i^{(pred)}$ based on $b'_{i-1}$ instead of $b_{i-1}$ as follows.

$$b_i^{(pred)} = \sigma\left(\sigma^{-1}(b'_{i-1}) + \Delta b_i\right),\tag{4}$$

Equation 4 is marked with green line in Fig. 4(b). Similarly, $b_{i+1}^{(pred)}$ is obtained as follows.

$$b_{i+1}^{(pred)} = \sigma\left(\sigma^{-1}(b'_i) + \Delta b_{i+1}\right) = \sigma\left(\sigma^{-1}(b'_{i-1}) + \Delta b_i + \Delta b_{i+1}\right)\tag{5}$$

Equation 5 is marked with red line in Fig. 4(b), where the gradients from $b_{i+1}^{(pred)}$ are propagated to $\theta_i$ through $\Delta b_i$.

Fig. 4 (c) shows a comparison of the performances of look forward once (LFO) and look forward twice (LFT) in different layers. For layer 0 to 2, LFO performs better than LFT. While LFT exceeds LFO in layer 3 to 6. This observation verifies our intuition that LFT sacrifices performance in early layers to achieve better final performance.

### 3.5 Mixed Query Selection

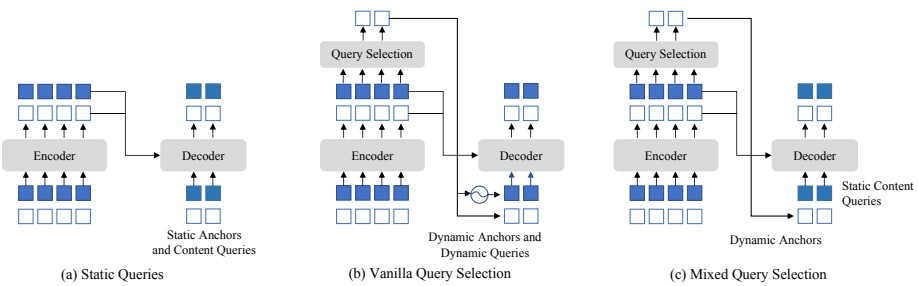

(a) Static Queries   (b) Vanilla Query Selection   (c) Mixed Query Selection

Figure 5: Comparison of three different query initialization methods. "static" means that queries will keep the same for different images in inference. A common implementation for these static queries is to make them learnable. Note that in (b) the selected reference points go through a positional encoding and linear transform to obtain the query embeddings as implemented in deformable DETR.

In DETR (Carion et al., 2020) and DN-DETR (Li et al., 2022), decoder queries are static embeddings without taking any encoder features from an individual image, as shown in Fig. 5 (a). They learn anchors or positional queries from training data and set the content queries as 0 vectors. Deformable DETR (Zhu et al., 2021) learns both the positional and content queries, which is another implementation of static query initialization. To further improve the performance, Deformable DETR (Zhu et al., 2021) has a query selection variant (or "two-stage"). It selects positions with top $K$ classification scores as reference points and the content queries are linear transform of the positional embeddings of the reference points. In addition, features in the selected positions go through a classification head and a box head to calculate auxiliary loss. We call the implementation in Deformable DETR as vanilla query selection as shown in Fig. 5. Vanilla query selection helps the model converge especially in early training epochs. However, its content queries are not aligned with those in CDN part—the content queries in CDN part are learnable class embeddings. Therefore, we propose to use selected positions as anchors and learnable query embeddings as the content queries. We call our method as mixed query selection. We show in Table 5 that our simple and intuitive method achieves better result.

### 4 Experiments

#### 4.1 Setup

**Dataset and Backbone:** We conduct evaluation on the COCO 2017 object detection dataset (Lin et al., 2014), which is split into `train2017` and `val2017` (also called `minival`). We report results with two different backbones: ResNet-50 (He et al., 2016) pre-trained on ImageNet-1k (Deng et al., 2009) and SwinL (Liu et al., 2021b) pre-trained on ImageNet-22k (Deng et al., 2009). DINO with ResNet-50 is trained on `train2017` without extra data, while DINO with SwinL is first pre-trained on Object365 (Shao et al., 2019) and then fine-tuned on `train2017`. We also report the `test-dev` results for DINO with SwinL.

**Implementation Details:** In appendix F, we provide implementation details, including all the hyperparameters and engineering techniques used in our models.

#### 4.2 Main Results

**12-epoch setting:** With our improved anchor box denoising and training losses, the training process can be significantly accelerated. As shown in Table 2, we compare our method with strong baselines

| Model | Epochs | AP | $AP_{50}$ | $AP_{75}$ | $AP_S$ | $AP_M$ | $AP_L$ | GFLOPS | Params | FPS |
|---|---|---|---|---|---|---|---|---|---|---|
| Faster-RCNN(5scale) Ren et al. (2015) | 12 | 37.9 | 58.8 | 41.1 | 22.4 | 41.1 | 49.1 | 207 | 40M | 21* |
| DETR(DC5) Carion et al. (2020) | 12 | 15.5 | 29.4 | 14.5 | 4.3 | 15.1 | 26.7 | 225 | 41M | 20 |
| Deformable DETR(4scale)Zhu et al. (2021) | 12 | 41.1 | – | – | – | – | – | 196 | 40M | 24 |
| DAB-DETR(DC5)† Liu et al. (2022) | 12 | 38.0 | 60.3 | 39.8 | 19.2 | 40.9 | 55.4 | 256 | 44M | 17 |
| Dynamic DETR(5scale) Dai et al. (2021b) | 12 | 42.9 | 61.0 | 46.3 | 24.6 | 44.9 | 54.4 | – | 58M | – |
| Dynamic Head(5scale) Dai et al. (2021a) | 12 | 43.0 | 60.7 | 46.8 | 24.7 | 46.4 | 53.9 | – | – | – |
| HTC(5scale) Chen et al. (2019a) | 12 | 42.3 | – | – | – | – | – | 441 | 80M | 5* |
| DN-Deformable-DETR(4scale)† Li et al. (2022) | 12 | 43.4 | 61.9 | 47.2 | 24.8 | 46.8 | 59.4 | 265 | 48M | 23 |
| DINO-4scale† | 12 | **49.0**(+5.6) | **66.6** | **53.5** | **32.0**(+7.2) | **52.3** | **63.0** | 279 | 47M | 24 |
| DINO-5scale† | 12 | **49.4**(+6.0) | **66.9** | **53.8** | **32.3**(+7.5) | **52.5** | **63.9** | 860 | 47M | 10 |

Table 2: Results for DINO and other detection models with the ResNet50 backbone on COCO `val2017` trained with 12 epochs (the so called $1\times$ setting). For models without multi-scale features, we test their GFLOPS and FPS for their best model ResNet-50-DC5. DINO uses 900 queries. † indicates models that use 900 queries or 300 queries with 3 patterns which has similar effect with 900 queries. Other DETR-like models except DETR (100 queries) uses 300 queries. * indicates that they are tested using the mmdetection Chen et al. (2019b) framework.

| Model | Epochs | AP | $AP_{50}$ | $AP_{75}$ | $AP_S$ | $AP_M$ | $AP_L$ |
|---|---|---|---|---|---|---|---|
| Faster-RCNN Ren et al. (2015) | 108 | 42.0 | 62.4 | 44.2 | 20.5 | 45.8 | 61.1 |
| DETR(DC5) Zhu et al. (2021) | 500 | 43.3 | 63.1 | 45.9 | 22.5 | 47.3 | 61.1 |
| Deformable DETR Zhu et al. (2021) | 50 | 46.2 | 65.2 | 50.0 | 28.8 | 49.2 | 61.7 |
| SMCA-R Gao et al. (2021) | 50 | 43.7 | 63.6 | 47.2 | 24.2 | 47.0 | 60.4 |
| TSP-RCNN-R Sun et al. (2020) | 96 | 45.0 | 64.5 | 49.6 | 29.7 | 47.7 | 58.0 |
| Dynamic DETR(5scale) Dai et al. (2021a) | 50 | 47.2 | 65.9 | 51.1 | 28.6 | 49.3 | 59.1 |
| DAB-Deformable-DETR Liu et al. (2022) | 50 | 46.9 | 66.0 | 50.8 | 30.1 | 50.4 | 62.5 |
| DN-Deformable-DETR Li et al. (2022) | 50 | 48.6 | 67.4 | 52.7 | 31.0 | 52.0 | 63.7 |
| DINO-4scale | 24 | **50.4**(+1.8) | 68.3 | 54.8 | 33.3 | 53.7 | 64.8 |
| DINO-5scale | 24 | **51.3**(+2.7) | 69.1 | 56.0 | 34.5 | 54.2 | 65.8 |
| DINO-4scale | 36 | **50.9**(+2.3) | 69.0 | 55.3 | 34.6 | 54.1 | 64.6 |
| DINO-5scale | 36 | **51.2**(+2.6) | 69.0 | 55.8 | 35.0 | 54.3 | 65.3 |

Table 3: Results for DINO and other detection models with the ResNet-50 backbone on COCO `val2017` trained with more epochs (24, 36, or more).

including both convolution-based methods (Ren et al., 2015; Chen et al., 2019a; Dai et al., 2021a) and DETR-like methods (Carion et al., 2020; Zhu et al., 2021; Dai et al., 2021b; Liu et al., 2022; Li et al., 2022). For a fair comparison, we report both GFLOPS and FPS tested on the same A100 NVIDIA GPU for all the models listed in Table 2. All methods except for DETR and DAB-DETR use multi-scale features. For those without multi-scale features, we report their results with ResNet-DC5 which has a better performance for its use of a dilated larger resolution feature map. Since some methods adopt 5 scales of feature maps and some adopt 4, we report our results with both 4 and 5 scales of feature maps.

As shown in Table 2, our method yields an improvement of $+5.6$ AP under the same setting using ResNet-50 with 4-scale feature maps and $+6.0$ AP with 5-scale feature maps. Our 4-scale model does not introduce much overhead in computation and the number of parameters. Moreover, our method performs especially well for small objects, gaining $+7.2$ AP with 4 scales and $+7.5$ AP with 5 scales.

**Comparison with the best models with a ResNet-50 backbone:** To validate the effectiveness of our method in improving both convergence speed and performance, we compare our method with several strong baselines using the same ResNet-50 backbone. Despite the most common 50-epoch setting, we adopt the 24 ($2\times$) and 36 ($3\times$) epoch settings since our method converges faster and yields only a smaller additional gain with 50-epoch training. The results in Table 3 show that, using only 24 epochs, our method achieves an improvement of $+1.8$ AP and $+2.7$ AP with 4 and 5 scales, respectively. Moreover, using 36 epochs in the $3\times$ setting, the improvement increases to $+2.3$ and $+2.6$ AP with 4 and 5 scales, respectively. The convergence curve comparison is shown in Fig. 6. We also show our results using SwinL backbone without bells and whistles in Appendix B.

### 4.3 COMPARISON WITH SOTA MODELS

To compare with SOTA results, we use the publicly available SwinL (Liu et al., 2021b) backbone pre-trained on ImageNet-22K. We first pre-train DINO on the Objects365 (Shao et al., 2019) dataset and then fine-tune it on COCO. As shown in Table 4, DINO achieves the best results of 63.2AP and 63.3AP on COCO `val2017` and `test-dev` with model size under 1 billion parameters, which demonstrate its strong scalability to larger model size and data size. Note that all the previous SOTA models in Table 4 do not use Transformer decoder-based detection heads (HTC++ (Chen et al., 2019a) and DyHead (Dai et al., 2021a)). It is the first time that an end-to-end Transformer detector is established as a SOTA model on the leaderboard (pap). Compared with the previous SOTA

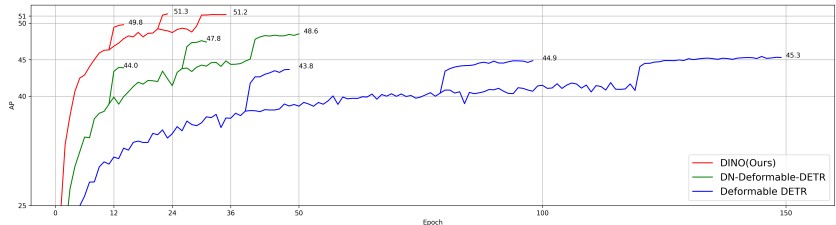

Figure 6: Training convergence curves evaluated on COCO `val2017` for DINO and two previous state-of-the-art models with ResNet-50 using multi-scale features.

| Method | Params | Backbone Pre-training Dataset | Detection Pre-training Dataset | Use Mask | End-to-end | val2017 (AP) | | test-dev (AP) | |
|---|---|---|---|---|---|---|---|---|---|
| | | | | | | w/o TTA | w/ TTA | w/o TTA | w/ TTA |
| SwinL Liu et al. (2021b) | 284M | IN-22K-14M | O365 | ✓ | | 57.1 | 58.0 | 57.7 | 58.7 |
| DyHead Dai et al. (2021a) | ≥ 284M | IN-22K-14M | Unknown* | | | – | 58.4 | – | 60.6 |
| Soft Teacher+SwinL Xu et al. (2021) | 284M | IN-22K-14M | O365 | ✓ | | 60.1 | 60.7 | – | 61.3 |
| GLIP Li et al. (2021) | ≥ 284M | IN-22K-14M | FourODs Li et al. (2021),GoldG+ Kamath et al. (2021) | | | – | 60.8 | – | 61.5 |
| Florence-CoSwin-HYuan et al. (2021) | ≥ 637M | FLD-900M Yuan et al. (2021) | FLD-9M Yuan et al. (2021) | | | – | 62.0 | – | 62.4 |
| SwinV2-G Liu et al. (2021a) | 3.0B | IN-22K-ext-70M Liu et al. (2021a) | O365 | ✓ | | 61.9 | 62.5 | – | 63.1 |
| DINO-SwinL(Ours) | **218M** | IN-22K-14M | O365 | | ✓ | **63.1** | **63.2** | **63.2** | **63.3** |

Table 4: Comparison of the best detection models on MS-COCO. Similar to DETR Carion et al. (2020), we use the term "end-to-end" to indicate if a model is free from hand-crafted components like RPN and NMS. The term "use mask" means whether a model is trained with instance segmentation annotations. We use the terms "IN" and "O365" to denote the ImageNet Deng et al. (2009) and Objects365 Shao et al. (2019) datasets, respectively. Note that "O365" is a subset of "FourODs" and "FLD-9M". * DyHead does not disclose the details of the datasets used for model pre-training.

models, we use a much smaller model size (1/15 parameters compared with SwinV2-G (Liu et al., 2021a)), backbone pre-training data size (1/60 images compared with Florence), and detection pre-training data size (1/5 images compared with Florence), while achieving better results. In addition, our reported performance without test time augmentation (TTA) is a neat result without bells and whistles. These results effectively show the superior detection performance of DINO compared with traditional detectors.

## 4.4 ABLATION

| #Row | QS | CDN | LFT | AP | $AP_{50}$ | $AP_{75}$ | $AP_S$ | $AP_M$ | $AP_L$ |
|---|---|---|---|---|---|---|---|---|---|
| 1. Optimized DN-Deformable DETR[†] Li et al. (2022) | No | | | 46.3 | 63.8 | 50.3 | 28.2 | 49.6 | 61.7 |
| 2. Row1+CDN* | No | ✓ | | 47.2 | 65.0 | 51.2 | 29.4 | 50.7 | 62.5 |
| 3. Row2+vanilla query selection Zhu et al. (2021) | Vanilla | ✓ | | 47.8 | 65.6 | 52.5 | 31.1 | 51.1 | 62.5 |
| 4. Row2+mixed query selection | Mixed | ✓ | | 48.6 | 66.0 | 52.9 | 31.3 | 51.9 | 62.7 |
| 5. DINO (ours, Row4+look forward twice) | Mixed | ✓ | ✓ | 49.0 | 66.6 | 53.5 | 32.0 | 52.3 | 63.0 |

Table 5: Ablation comparison of the proposed algorithm components. We use the terms "QS", "CDN", and "LFT" to denote "Query Selection", "Contrastive De-Noising Training", and "Look Forward Twice", respectively. [†] We propose an optimized DN-Deformable DETR with our technical improvements. The technical details are shown in Appendix A. * We also use adaptive number of denoising groups here.

**Effectiveness of New Algorithm Components:** We validate the effectiveness of our proposed methods in Table 5. We build an optimized DN-Deformable DETR as our strong baseline, which performs better than the one in Table 2. We include all the pipeline optimization and engineering techniques (see section 4.1 and Appendix F) in the strong baseline. The result of the strong baseline is available in Table 5 Row 1. According to Table 5, our three new methods in DINO further improve the performance significantly even without considering any engineering techniques.

## 5 CONCLUSION

In this paper, we have presented a strong end-to-end Transformer detector DINO with contrastive denoising training, look forward twice, and mixed query selection, which significantly improves both the training efficiency and the final detection performance. As a result, DINO outperforms all previous ResNet-50-based models on COCO `val2017` in both the 12-epoch and the 36-epoch settings using multi-scale features. Motivated by the improvement, we further explored to train DINO with a stronger backbone on a larger dataset and achieved a strong result, 63.3 AP on COCO 2017 `test-dev`. This result establishes DETR-like models as a mainstream detection framework, not only for its novel end-to-end detection optimization, but also for its superior performance.

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

# A OPTIMIZED DN-DEFORMABLE DETR

The optimized DN-Deformable DETR differs from the original DN-Deformable DETR in the following three parts. Firstly, the optimized DN-Deformable DETR adopts deformable attention in both encoder and decoder while the original one only adopts deformable attention in encoder. With deformable attention in decoder, the optimized one is able to use more decoder queries. For example, we use 900 here. Secondly, the optimized one use different weight for matcher and loss, while the original one follows DETR to use same weight for loss and matcher. For example, we use class weight 1.0 for loss and 2.0 for matcher. Finally, we set dropout rate to be 0. We find these three technical improvements can improve the performance.

# B RESULTS USING SWINL BACKBONE WITHOUT PRE-TRAINING ON OBJECT 365

We also evaluate our method on COCO `val2017` with SwinL as backbone without pre-training on Object 365. The results are without any bells and whistles. We compare with other methods using Swin-L backbone.

| Model | Epochs | AP | $AP_{50}$ | $AP_{75}$ | $AP_S$ | $AP_M$ | $AP_L$ |
|---|---|---|---|---|---|---|---|
| Cascade Mask RCNN-SwinL Cai & Vasconcelos (2018) | – | 55.0 | – | – | – | – | – |
| HTC++-SwinL Chen et al. (2019a) | – | 57.1 | – | – | – | – | – |
| DINO-4scale-SwinL | 36 | **58.0** | **76.7** | **63.4** | **41.3** | **61.9** | **73.7** |
| DINO-5scale-SwinL | 36 | **58.5** | **77.0** | **64.1** | **41.5** | **62.3** | **74.0** |

Table 6: Results for DINO and other detection models with the SwinL backbone on COCO `val2017`.

# C TEST TIME AUGMENTATIONS (TTA)

We aim to build an end-to-end detector that is free from hand-crafted components. However, to compare with traditional detection models, we also explore the use of TTA in DETR-like models. We only use it in our large model with the SwinL backbone. Our TTA does not obtain an inspiring gain compared with traditional detectors, but we hope our exploration may provide some insights for future studies.

We adopt multi-scale test and horizontal flip as TTA. However, the way of ensembling different augmentations in our method is different from that in traditional methods which usually output duplicate boxes. In traditional methods, the ensembling is done by first gathering predictions from all augmentations and ranked by a confidence score. Then, duplicate boxes are found and eliminated by NMS or box voting. The reason why predictions from all augmentations are gathered first is that duplicate boxes appear not only among different augmentations but also within one augmentation. This ensembling method decreases the performance for our method since DETR-like methods are not prone to output duplicate boxes since their set-based prediction loss inhibits duplicate predictions and ensembling may incorrectly remove true positive predictions (Carion et al., 2020). To address this issue, we designed a one-to-one ensembling method. Assume we have $n$ augmentations $Aug_0, Aug_1, ..., Aug_{n-1}$, where $Aug_i$ has predictions $\mathbf{O}^i$ and a pre-defined hyper-parameter weight $w^i$. $\mathbf{O}^i = \left\{ (b_0^i, l_0^i, s_0^i), (b_1^i, l_1^i, s_1^i), ..., (b_{m-1}^i, l_{m-1}^i, s_{m-1}^i) \right\}$ where $b_j^i, l_j^i$ and $s_j^i$ denote the $j$-th boundbox, label and score, respectively. We let $Aug_0$ be the main augmentation which is the most reliable one. For each prediction in $\mathbf{O}^0$, we select the prediction with the highest $IOU$ from predictions of each of other augmentations $\mathbf{O}^1, ..., \mathbf{O}^{n-1}$ and make sure the $IOU$ is higher than a predefined threshold. Finally, we ensemble the selected boxes through weighted average as follows

$$b = \frac{1}{\sum I^i} \sum_{i=o}^{n-1} I^i w^i s_{idx(i)}^i b_{idx(i)}^i \qquad (6)$$

where $I^i = 1$ when there is at least one box in $\mathbf{O}^i$ with IOU higher than the threshold and $I^i = 0$ otherwise. $idx(i)$ denotes the index of the selected box in $\mathbf{O}^i$.

# D    TRAINING EFFICIENCY

We provide the GPU memory and training time for our base model in Table 7. All results are reported on 8 Nvidia A100 GPUs with ResNet-50 (He et al., 2016). The results demonstrate that our models are not only effective but also efficient for training.

| Model | #images per GPU | Traning Time | GPU Mem. | Epoch | AP |
|---|---|---|---|---|---|
| Faster RCNN (Ren et al., 2015)* | 8 | $\sim$ 60min/ep | 13GB | 108 | 42.0 |
| DETR (Carion et al., 2020) | 8 | $\sim$ 16min/ep | 26GB | 300 | 41.2 |
| Deformable DETR (Zhu et al., 2021)* | 2 | $\sim$ 55min/ep | 16GB | 50 | 45.4 |
| DINO(Ours) | 2 | $\sim$ 55min/ep | 16GB | **12** | **49.0** |

Table 7: Training efficieny for different models with ResNet-50 backbone. All models are trianed with 8 Nvidia A100 GPUs. All results are reported by us. * The results of Faster RCNN are tested with the mmdetection framework. * We use the vanilla Deformable DETR without two-stage and bbox refinement during testing.

# E    ADDITIONAL ANALYSIS ON OUR MODEL COMPONENTS

| # Encoder/Decoder | 6/6 | 4/6 | 3/6 | 2/6 | 6/4 | 6/2 | 2/4 | 2/2 |
|---|---|---|---|---|---|---|---|---|
| AP | 47.4 | 46.2 | 45.8 | 45.4 | 46.0 | 44.4 | 44.1 | 41.2 |

Table 8: Ablation on the numbers of encoder layers and decoder layers with the ResNet-50 backbone on COCO `val2017`. We use the 12-epoch setting and 100 DN queries without negative samples here.

**Analysis on the Number of Encoder and Decoder Layers:** We also investigate the influence of varying numbers of encoder and decoder layers. As shown in Table 8, decreasing the number of decoder layers hurts the performance more significantly. For example, using the same 6 encoder layers while decreasing the number of decoder layers from 6 to 2 leads to a 3.0 AP drop. This performance drop is expected as the boxes are dynamically updated and refined through each decoder layer to get the final results. Moreover, we also observe that compared with other DETR-like models like Dynamic DETR (Dai et al., 2021a) whose performance drops by 13.8AP (29.1 vs 42.9) when decreasing the number of decoder layers to 2, the performance drop of DINO is much smaller. This is because our mixed query selection approach feeds the selected boxes from the encoder to enhance the decoder queries. Therefore, the decoder queries are well initialized and not deeply coupled with decoder layer refinement.

| # Denoising query | 100 CDN | 1000 DN | 200 DN | 100 DN | 50 DN | 10 DN | No DN |
|---|---|---|---|---|---|---|---|
| AP | 47.9 | 47.6 | 47.4 | 47.4 | 46.7 | 46.0 | 45.1 |

Table 9: Ablation on number of denoising queries with the ResNet-50 backbone on COCO validation. Note that 100 CND query pairs contains 200 queries which are 100 positive and 100 negative queries.

**Analysis on Query Denoising:** We continue to investigate the influence of query denoising by varying the number of denoising queries. We use the optimized dynamic denoising group (detailed in Appendix F.1). As shown in Table 9, when we use less than 100 denoising queries, increasing the number can lead to a significant performance improvement. However, continuing to increase the DN number after 100 yields only a small additional or even worse performance improvement. We also analysis the effect of the number of encoder and decoder Layers in Appendix E.

## F  More Implementation Details

### F.1  Adaptive DN groups

In DN-DETR, all the GT objects (label+box) in one image are collected as one GT group for denoising. To improve the DN training efficiency, multiple noised versions of the GT group in an image are used during training. In DN-DETR, the number of groups is set to five or ten according to different model sizes. As DETR-like models adopt mini-batch training, the total number of DN queries for each image in one batch is padded to the largest one in the batch. Considering that the number of objects in one image in COCO dataset ranges from 1 to 80, this design is inefficient and results in excessive memory consumption. To address this problem, we propose to fix the number of DN queries and dynamically adjust the number of groups for each image according to its number of objects.

### F.2  Large-Scale Model Pre-trianing

Objects365 (Shao et al., 2019) is a large-scale detection data set with over $1.7M$ annotated images for training and $80,000$ annotated images for validation. To use the data more efficiently, We select the first $5,000$ out of $80,000$ validation images as our validation set and add the others to training. We pre-train DINO on Objects365 for 26 epochs using 64 Nvidia A100 GPUs and fine-tune the model on COCO for 18 epochs using 16 Nvidia A100 GPUS. Each GPU has a local batch size of 1 image only. In the fine-tuning stage, we enlarge the image size to $1.5\times$ (i.e., with max size $1200 \times 2000$). This adds around $0.5$ AP to the final result. To reduce the GPU memory usage, we leverage checkpointing (Chen et al., 2016) and mixed precision (Micikevicius et al., 2018) during training. Moreover, we use 1000 DN queries for this large model.

### F.3  Other Implementation Details

#### F.3.1  Basic hyper-parameters.

For hyper-parameters, as in DN-DETR, we use a 6-layer Transformer encoder and a 6-layer Transformer decoder and 256 as the hidden feature dimension. We set the initial learning rate (lr) as $1 \times 10^{-4}$ and adopt a simple lr scheduler, which drops lr at the 11-th, 20-th, and 30-th epoch by multiplying 0.1 for the 12, 24, and 36 epoch settings with RestNet50, respectively. We use the AdamW (Kingma & Ba, 2014; Loshchilov & Hutter, 2017) optimizer with weight decay of $1 \times 10^{-4}$ and train our model on Nvidia A100 GPUs with batch size 16. Since DN-DETR (Li et al., 2022) adopts 300 decoder queries and 3 patterns (Wang et al., 2021), we use $300 \times 3 = 900$ decoder queries with the same computation cost. Learning schedules of our DINO with SwinL are available in the appendix.

#### F.3.2  Loss function.

We use the L1 loss and GIOU (Rezatofighi et al., 2019) loss for box regression and focal loss (Lin et al., 2020) with $\alpha = 0.25, \gamma = 2$ for classification. As in DETR (Carion et al., 2020), we add auxiliary losses after each decoder layer. Similar to Deformable DETR (Zhu et al., 2021), we add extra intermediate losses after the query selection module, with the same components as for each decoder layer. We use the same loss coefficients as in DAB-DETR (Liu et al., 2022) and DN-DETR (Li et al., 2022), that is, 1.0 for classification loss, 5.0 for L1 loss, and 2.0 for GIOU loss.

#### F.3.3  Detailed model components.

We also optimize the detection pipeline used in DAB-DETR (Liu et al., 2022) and DN-DETR (Li et al., 2022). Following DN-Deformable-DETR (Li et al., 2022), we use the same multi-scale approach as in Deformable DETR (Zhu et al., 2021) and adopt the deformable attention. DN-DETR uses different prediction heads with unshared parameters in different decoder layers. In addition, we introduce dynamic denoising group to increase denoising training efficiency and alleviate memory overhead (see Appendix F.1). In this work, we find that using a shared prediction head will add additional performance improvement. This also leads to a reduction of about one million parameters.

| Item | Value |
|---|---|
| lr | 0.0001 |
| lr_backbone | 1e-05 |
| weight_decay | 0.0001 |
| clip_max_norm | 0.1 |
| pe_temperature | 20 |
| enc_layers | 6 |
| dec_layers | 6 |
| dim_feedforward | 2048 |
| hidden_dim | 256 |
| dropout | 0.0 |
| nheads | 8 |
| num_queries | 900 |
| enc_n_points | 4 |
| dec_n_points | 4 |
| transformer_activation | "relu" |
| batch_norm_type | "FrozenBatchNorm2d" |
| set_cost_class | 2.0 |
| set_cost_bbox | 5.0 |
| set_cost_giou | 2.0 |
| cls_loss_coef | 1.0 |
| bbox_loss_coef | 5.0 |
| giou_loss_coef | 2.0 |
| focal_alpha | 0.25 |
| dn_box_noise_scale | 0.4 |
| dn_label_noise_ratio | 0.5 |

Table 10: Hyper-parameters used in our models.

In addition, we find the conditional queries (Meng et al., 2021) used in DAB-DETR does not suit our model and we do not include them in our final model.

### F.3.4 TRAINING AUGMENTATION.

We use the same random crop and scale augmentation during training following DETR (Carion et al., 2020). For example, we randomly resize an input image with its shorter side between $480$ and $800$ pixels and its longer side at most $1333$. For DINO with SwinL, we pre-train the model using the default setting, but finetune using $1.5\times$ larger scale (shorter side between $720$ and $1200$ pixels and longer side at most $2000$ pixels) to compare with models on the leaderboard (pap). Without using any other tricks, we achieve the result of $63.1$ on `val2017` and $63.2$ on `test-dev` without test time augmentation (TTA) (see Appendix C), outperforming the previous state-of-the-art result $63.1$ achieved by SwinV2 (Liu et al., 2021a) with a much neater solution.

### F.3.5 MULTI-SCALE SETTING.

For our $4$-scale models, we extract features from stages 2, 3, and $4$ of the backbone and add an extra feature by down-sampling the output of the stage $4$. An additional feature map of the backbone stage 1 is used for our 5-scale models. For hyper-parameters, we set $\lambda_1 = 1.0$ and $\lambda_2 = 2.0$ and use $100$ CDN pairs which contain $100$ positive queries and $100$ negative queries.

### F.4 DETAILED HYPER-PARAMETERS

We list the hyper-parameters for those who want to reproduce our results in Table 10.

|  | #parameters | GFLOPs | FPs |
|---|---|---|---|
| DINO-Swin-L-4Scale | 217.6 | 1284.5 | 8.1 |
| DINO-Swin-L-5Scale | 217.2 | 703.5 | 12.8 |

Table 11: The inference speed and computation cost of our laege model.

|  | # epochs | AP | $AP_{50}$ | $AP_{75}$ | $AP_S$ | $AP_M$ | $AP_L$ |
|---|---|---|---|---|---|---|---|
| DINO-LFT | 12 | 49.0 | 66.6 | 53.5 | 32.0 | 52.3 | 63.0 |
| DINO-LF3 | 12 | 48.3 | 65.5 | 52.7 | 31.2 | 51.7 | 62.5 |
| DINO-LF4 | 12 | 48.2 | 65.4 | 52.9 | 30.4 | 51.8 | 62.9 |

Table 12: The experiments of Look Forward Three (LF3) and Four times (LF4).

## G   INFERENCE SPEED AND GFLOPS

We list the inference cost of our 4-scale and 5-scale model with Swin-L backbones in Table 11. Note that our model for Table 4 is a 5-scale model.

## H   WHY DINO IMPROVES AP ON SMALL OBJECTS BY LARGE

There are several reasons for the large AP improvement on small objects ($AP_s$).

1. In Table 5, our optimized DN-Deformable DETR has $AP_s$ of 28.2 which is +3.4 higher than that of the original DN-Deformable DETR in Table 2. The original one uses dense attention in the decoder while the optimized one uses deformable attention which is better for local attention and therefore improves $AP_s$. In addition, we fixed a problem in the original DN-Deformable DETR's Transformer encoder—their deformable attention is not properly initialized using the initialization method in Deformable DETR. The Transformer encoder is a critical component that processes multi-scale image features and multi-scale image features are critical for small objects. Therefore, the optimized one has higher $AP_s$.

2. In Table 5, we can see that CDN improves $AP_s$ by 1.2. By introducing negative noised queries, CDN encourages the model to pick up the anchor nearest to the center of a GT box to make predictions and explicitly suppresses farther anchors. Since small object detection is more sensitive to the quality of anchors, DINO with high-quality anchors can achieve better $AP_s$.

3. Query selection improves $AP_s$ by 1.9. Query selection provides high-quality anchor initialization which is especially beneficial to small objects. The reason is similar to reason 2 that small object detection is more sensitive to the quality of anchors.

## I   MORE DETAILS ABOUT LOOK FORWARD TWICE (LFT)

We propose LFT because the original Look Forward Once scheme for box refinement is greedy, which will lead to sub-optimal results. We propose LFT to make the model far-sighted. But there is a trade-off. When we increase the number of layers to "look forward", the model becomes harder to converge. We conduct experiments of Look Forward Three and Four times as shown in Table 12. The results become worse when we continue to increase the number of layers to "look forward".

Following is a detailed explanation of why LFT is worse than LFO in layers 0 to 2 but outperforms LFO in layers 3 to 6 as shown in Fig. 4.

There are actually three factors affecting the performance of layer $i$.

1. The performance of layer $i - 1$. Since the predictions of layer $i$ are based on predictions of layer $i - 1$, better predictions in layer $i - 1$ lead to better predictions in layer $i$.

2. Whether allows gradients to backpropagate from layer $i$ to layer $i-1$. Allowing the gradient to propagate to layer $i - 1$ helps performance in layer $i$.

3. Whether allows gradients backpropagate from layer $i + 1$ to layer $i$. Allowing gradient to propagate from layer $i + 1$ to layer $i$ jeopardizes performance in layer $i$.

For layer 0, there is no layer $i - 1$, so factors 1 and 2 do not affect the performance. According to factor 3, LFO is better than LFT.

For layers 1 to 5, LFO has an advantage in factor 3 and LFT has an advantage in factor 2. Because factor 2 affects the performance more than factor 3, The gap between LFT is narrowed down from layer 0 to 2 and LFT exceeds LFO in layer 3.

In the last layer (when $i = 6$), there is no layer $i + 1$ (factor 3 does not affect the result) and LFT has the advantage in both factors 1 and 2. Therefore, LFT exceeds LFO in the last layer.

## J    VISUALIZATIONS

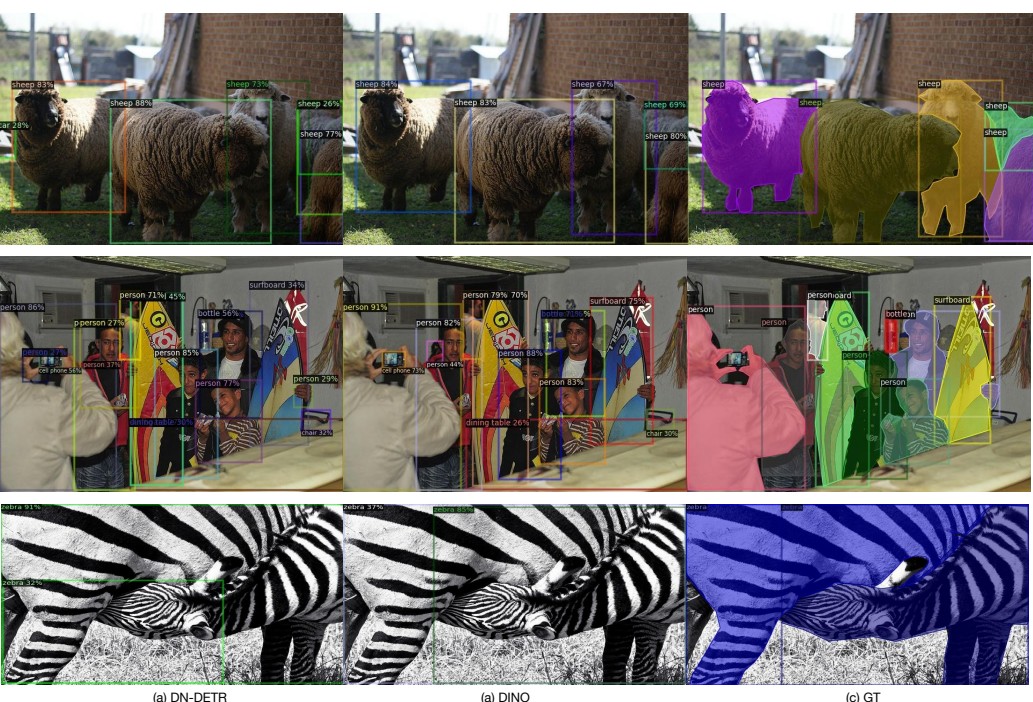

|          |         |        |
| :------: | :-----: | :----: |
| (a) DN-DETR | (a) DINO | (c) GT |

Figure 7: Visualization of cases DINO outperforms DN-DETR.

We present a comparison of visualizations in Fig. 7. The results show that our DINO has better predictions than DN-DETR.

## K    LVIS RESULTS

To evaluate DINO's performance on other detection datasets, we conducted experiments on more challenging LVIS (Gupta et al., 2019) dataset.

| Model | Backbone | Val v1.0 | | | |
|---|---|---|---|---|---|
| | | AP | $AP_r$ | $AP_c$ | $AP_f$ |
| Supervised-RFS (Gupta et al., 2019) | R50 | 25.4 | 12.3 | 24.3 | 32.4 |
| MaskRCNN-LOCE (He et al., 2017) | R50 | 27.4 | - | - | - |
| GLIP*(Li et al., 2021) | Swin-L | 26.9 | 17.1 | 23.3 | 35.4 |
| DINO[†] | R50 | 31.2 | 24.5 | 29.8 | 35.7 |

Table 13: The results on LVIS val v1.0. * denotes zero-shot results. [†] denotes DINO is trained for 12 epochs and is not fully conveged.

