# OpenReview forum: "DINO: DETR with Improved DeNoising Anchor Boxes for End-to-End Object Detection"
_ICLR.cc/2023/Conference — ICLR 2023 poster_

### Official Review · Reviewer_tCi7 · 2022-10-24

**Confidence:** 4
**Correctness:** 4
**Technical Novelty And Significance:** 3
**Empirical Novelty And Significance:** 3
**Recommendation:** 8

**Clarity, Quality, Novelty And Reproducibility:**

Unclarity and Questions:
1. The denoising step is actually to add many-to-one matching (many queries are assigned to one ground true bounding box) in the vanilla DETR-like detector training scheme that normally is a one-to-one on bipartite matching. Such a many-to-one matching strategy with contrastive training is commonly used in classical detectors, and NMS is needed in those detectors due to duplicated detections. Why author claim that explicitly introduces negative examples in denoising step for DETR-like models can reduce duplicate predictions, any differences between classical and DETR-like detectors in many-to-one matching and contrastive training?

2. Why making the content queries learnable can force the first decoder layer to focus on the spatial prior? As I though the behind reason of learnable scheme is to align with queries in CDN part. Please provide more explanations.

3. For a look forward twice technic, why LFO performs better than LFT in layer 0 to 2, while preforms pool in layer 3 to 6? And how the variants of looking forward trice or quartic perform in DETR-like models？


**Strength And Weaknesses:**

Strengths:
1) DINO points out the reason why newly developed DETR-like models are inferior to the classical detectors and propose the corresponding solutions, eg., denoising training, query initialization and box prediction to highly optimize DETR-series detectors.
2) The contrastive denoising training helps the model to predict more precise boxes and avoid duplicate outputs of the same target compared with DN-DETR.
3) The look forward twice scheme has the ability to correct the updated parameters with gradients from later layers to overcome the weakness of refining boxes in each decoder layer, while keeping the advantages of fast convergence.
4) Authors report the performance of DETR-like model on large backbone with large-scale pre-training data.

Weaknesses:

Please see the following part of `Unclarity and Questions'.




**Summary Of The Paper:**

In order to speed up the training convergence and highly optimize the Transformer-based detector (DETR-like models), a contrastive way for denoising training, a look forward twice scheme for box prediction, and a mixed query selection method for anchor initialization are proposed. The extensive ablation studies demonstrate the effectiveness of proposed methos over current DETR-like models.

**Summary Of The Review:**

I would tend to accept this paper as it is practical DETR-like model and the proposed modules are supported by empirical experiments.

---

> ### Author Response · Authors · 2022-11-13
> **Responses to the Reviewer tCi7**
>
> Thanks for your valuable questions about our paper. Your questions help us to get a deep understanding of our model design. We will respond to your concerns item by item in this section.
>
> ## 1.The relationship between denoising and many-to-one matching.
> Thank you for your valuable question, which brings us a new view of our denoising modules. We want to respond to the question in three folds.
> First, we think that our model is still a one-to-one matching method, as we assign each ground truth with one query during the matching part, with a bipartite matching algorithm. The denoising part introduces extra positive samples, which is a plug-in module for faster convergence. Hence **our model does not need NMS** as well.
> Then we want to compare our method with classical detectors. Although classic detectors have a process to assign positive and negative samples, which in some sense is a contrastive way, the assignment is performed at the proposed regions from RPN. It corresponds to our matching part, where we assign positive and negative samples with bipartite matching. However, our CDN process introduces extra ground-truth boxes with noises, which is not presented in most classical detectors.
> At last, we want to re-clarify the influence of duplicate predictions. Duplicate predictions occur when multiple queries (anchors) aim at one object. These anchors predicting the same objects are all near the object. In our CDN, the negative examples are designed to be close to GT. So it requires models to distinguish positive and negative ones. Our CDN helps the model learn to choose the nearest anchor and reject others and therefore helps reduce duplicate predictions.
>
> ## 2.The learnable content queries
> Thanks for your valuable questions. We are sorry that we have not made it clear in our initial submission. We hope that the "learnable scheme is to align with queries in CDN part" is our motivation and "force the first decoder layer to focus on the spatial prior" is another reason because we find that when we construct CDN queries, the content part is not important and the model focus more on spatial information
>
> ## 3. Why LFO performs better than LFT in layers 0 to 2, while preforms poor in layers 3 to 6:
> Thanks for your valuable questions. We are sorry we have not explained it clearly. Following we give a detailed analysis of the effect of different numbers to "look forward".
> There are three factors affecting the performance of layer i.
> 1. The performance of layer i-1. Since the predictions of layer i are based on predictions of layer i-1, better predictions in layer i-1 lead to better predictions in layer i.
> 2. Whether allow gradients backpropagate from layer i to layer i-1. Allowing the gradient to propagate to layer i-1 helps performance in layer i.
> 3. Whether allow gradients backpropagate from layer i+1 to layer i. Allowing gradient to propagate from layer i+1 to layer i jeopardizes performance in layer I.
>
> For layer 0, there is no layer i-1, so factors 1 and 2 do not affect the performance. According to factor 3, LFO is better than LFT.
>  For layers 1 to 5, LFO has an advantage in factor 3, and LFT has an advantage in factor 2. Because factor 2 affects the performance more than factor 3, The gap between LFT is narrowed down from layer 0 to 2, and LFT exceeds LFO in layer 3.
> Moreover, in the last layer (when i=6), there is no layer i+1（factor 3 does not affect the result）, and LFT has the advantage in both factors 1 and 2. Therefore LFT undoubtedly exceeds LFO.
> We have also put this explanation in the Appendix. We hope this can help readers to understand our LFT. Feel free to give us feedback if you are still confused.
>
> ## 4. How look forward trice or quartic performs?
> Thanks for your helpful suggestions. We conducted experiments of look forward trice and quartic these days and show the results as follows. The results show that twice is the best choice for our DINO model empirically. We also put the results in the revised paper Appendix Table 12.
> |          | # epochs | AP   | AP$_{50}$ | AP$_{75}$ | AP$_S$ | AP$_M$ | AP$_L$ |
> |----------|----------|------|-----------|-----------|--------|--------|--------|
> | DINO-LFT | 12       | 49.0 | 66.6      | 53.5      | 32.0   | 52.3   | 63.0   |
> | DINO-LF3 | 12       | 48.3 | 65.5      | 52.7      | 31.2   | 51.7   | 62.5   |
> | DINO-LF4 | 12       | 48.2 | 65.4      | 52.9      | 30.4   | 51.8   | 62.9   |
>
> We thank you again for your detailed comments and valuable questions. We hope our work can serve as a strong baseline for DETR-based models and help researchers to understand DETR-based detectors better.

---

### Official Review · Reviewer_dnJ7 · 2022-10-24

**Confidence:** 5
**Correctness:** 4
**Technical Novelty And Significance:** 2
**Empirical Novelty And Significance:** Not applicable
**Recommendation:** 8

**Clarity, Quality, Novelty And Reproducibility:**

The paper is clearly written, all the changes are well described, motivated, and empirically demonstrated.
I believe all the technical details provided, including hyper-parameters, are sufficient for reproduction.


Typo:
- "LFT scarifies performance" -> sacrifices

**Strength And Weaknesses:**

The paper shows strong results on the Coco benchmark, surpassing current SOTA even with less training data, which is an impressive feat.
I find all the comparisons to be fair with previous approaches.

On the presentation side, one minor weakness is the lack of visualization of any kind. For example, I would have liked to see some head-to-head inference comparison with DN-DETR on the same images, as well as some failure cases.

Finally, although it is not required for acceptance, I think the paper could be strengthen by showing results on datasets beyond Coco, perhaps LVIS or CrowdHuman.



**Summary Of The Paper:**

The paper proposes some improvements on the DETR family of Detectors.
Namely, the improvements are as follows:
- Add some negatives in the denoising objective
- Refine the design of the iterative refinement of boxes
- Propose a slightly different way to initialize object queries

With these improvements, the resulting model improves across the board on the Coco dataset, and in particular sets a new SOTA result for the setting with extra-data

**Summary Of The Review:**

The paper is technically sound and the results are strong, I recommend acceptance.

---

> ### Author Response · Authors · 2022-11-13
> **Responses to the Reviewer dnJ7**
>
> Thanks for your positive feedback on our strong results and fair comparison. Also thank you for your advice on writing and experiments. You have several suggestions about our work. We respond to your suggestions item by item in this section.
> ## 1. Suggestions on visualization:
> Thank you for your valuable suggestion, which helps us to improve our paper.. Visualization is important for presenting research works and conveying ideas, especially in computer vision. We have put some head-to-head inference comparisons with DN-DETR in Appendix J.
> ## 2. Suggestions on experiments on other datasets:
> Thank you for your valuable suggestions. This is actually what we plan to do in future work. These days we conducted an experiment on LVIS. The experiment will be finished in one or two days. We will post our results after finishing. Thank you again for your suggestion.
> ## 3. Our typo of "sacrifices":
> Thank you for your careful reading. We fixed the typo in the revised version.
>
> Thank you again for your helpful suggestions. We believe these suggestions help to improve the quality of our paper.

---

> > ### Comment · Reviewer_dnJ7 · 2022-11-16
> > **Thank you for the rebuttal**
> >
> > I thank the author for their thorough rebuttal. I maintain my recommendation of accepting this paper.

---

> > > ### Author Response · Authors · 2022-11-18
> > > **Tank you for keeping score**
> > >
> > > Thank you for keeping score and thank you again for your valuable advice.

---

> ### Author Response · Authors · 2022-11-16
> **Responses to the Reviewer dnJ7-LVIS results**
>
> Thank you for your suggestion to evaluate DINO on other datasets. We have run DINO for 12 epochs on LVIS. We show the results as follows and post the results to the Appendix K of the revised paper.
> Note that our model is trained only for 12 epochs and is probably  not fully conveged.  Note that GLIP results are zero-shot results for reference because thery only reported zero-shot results.
> | Model          | Backbone | Val  v1.0 |        |        |        |
> |----------------|:--------:|:---------:|--------|--------|--------|
> |                |          | AP        | AP$_r$ | AP$_c$ | AP$_f$ |
> | Supervised-RFS | R50      | 25.4      | 12.3   | 24.3   | 32.4   |
> |  MaskRCNN-LOCE | R50      | 27.4      | -      | -      | -      |
> | GLIP       | Swin-L   | 26.9      | 17.1   | 23.3   | 35.4   |
> | DINO    | R50      | 31.2      | 24.5   | 29.8   | 35.7   |

---

### Official Review · Reviewer_7vU5 · 2022-10-24

**Confidence:** 4
**Correctness:** 3
**Technical Novelty And Significance:** 2
**Empirical Novelty And Significance:** 2
**Recommendation:** 5

**Clarity, Quality, Novelty And Reproducibility:**

The quality of the paper is good: it studies the fundamental task, object detection in computer vision, and has achieved the new state-of-the-art results.

The clarity of the paper is good: introduction of the method, experiments and claims are clear enough to read and understand.

The novelty of the paper is *not* significant, since the three new techniques are the extensions or modifications of existing works.

Source code isn't provided but the authors state the code will be made available for reproducibility.


**Strength And Weaknesses:**

Strength:

The proposed method is simple yet effective.
The paper is well organized. The illustration provides the great details of the new techniques, which are helpful for the readers to understand.
The work has achieved strong experimental results with the improvement of about 2% over the state-of-the-art baselines.
The paper presents extensive details of the experiments.


Weaknesses

The motivation of the look forward twice scheme isn't very convincing. The paper has shown that the choice of twice is better than once, but is it better than three times and four times? If there is no good theoretical analysis to support this choice, it would be better to have some empirical results to justify it.

The second last sentence in ABSTRACT, “…, DINO significantly reduces its model size and pre-training data size while achieving better results.”  The claim is confusing. To my understanding, with the training techniques of DINO, the model with small size can achieve better performance than the large models. But the approach of DINO itself does not reduce the model size.

In section 3.3, a typo in “They are are expected to predict “no object”.

**Summary Of The Paper:**

This paper presents an improved DETR-like algorithm (both accuracy and speed) for object detection. Three techniques are proposed in the paper, including contrastive denoising training, look forward twice, and mixed query selection. The idea of denoising anchor boxes in previous work has shown effectiveness in stabilizing training and accelerating convergence. Based on the previous work, the authors introduce the additional hard negative examples to further improve the performance. The experimental results of the 12-epoch setting show that the new contrastive denoising technique leads to a faster convergence. Furthermore, the ablation study shows that this new technique has achieved the gain of about 1% in detection accuracy. The Look Forward Twice technique for iterative box refinement is the extension of the look forward once in previous work Deformable DETR. The backpropagation of the gradient is performed in two consecutive decoder layers instead of one layer. The ablation study shows that this technique boosts the performance by about 0.5%. The Mixed Query Selection technique is a combination of previous works DN-DETR and Deformable DETR, with selected positions anchors and learnable the content queries. The ablation study also shows that this technique boosts the performance by about 0.5%.

Furthermore, the experimental results show that the performance of the model with about 1 billion parameters trained by the new approach outperforms the larger models trained from even more pre-training data.


**Summary Of The Review:**

I would recommend a marginally below acceptance. The novelty of this work is not significant as the three techniques are quite incremental without theoretical insights. The experimental results and reported performances are the state-of-the-arts and may have a good impact to the field of objection detection.

---

> ### Author Response · Authors · 2022-11-13
> **Responses to Reviewer 7vU5 part 1**
>
> Thanks for your positive feedback on our paper writing and experiments. We also appreciate your advice on writing. We respond to your concerns item by item in this section.
> ## 1. Concerns about the motivation of LFT:
> Thank you for your question. We do experiments on looking forward three and four times and show the results as follows.
> | Model | # epochs | AP   | AP$_{50}$ | AP$_{75}$ | AP$_S$ | AP$_M$ | AP$_L$ |
> |-------|----------|------|-----------|-----------|--------|--------|--------|
> | DINO-LFT |   12     | 49.0 | 66.6      | 53.5      | 32.0   | 52.3   | 63.0   |
> | DINO-LF3 |   12     | 48.3 | 65.5      | 52.7      | 31.2   | 51.7   | 62.5   |
> | DINO-LF4 |   12     | 48.2 | 65.4      | 52.9      | 30.4   | 51.8   | 62.9   |
>
> The empirical results show that looking forward twice achieves the best result. There is a trade-off between the model field of view and the optimization. With the number of layers to "look forward" increasing, the model becomes more far-sighted, yet it is also harder to converge because a layer receives gradients from more other layers. We also put our results of look forward three and four times in the revised paper Appendix Table 12. In addition, we also put an explanation of why LFT is worse than LFO in layers 0 to 2 and exceeds LFO in layers 3 to 6 in Appendix Sec. H.
> ## 2. Concerns about the sentence in the abstract:“…, DINO significantly reduces its model size and pre-training data size while achieving better results.”
> Thank you for your careful reading and valuable suggestions. Your suggestion helps to make the paper clearer. We have modified it as "DINO achieves better results with smaller model size and pre-training data size". We hope this can address your concern.
> ## 3.The typo in Sec. 3:
> Thank you for your careful reading. We have fixed the typo in the revised paper.
>
> Thanks again for your careful reading and valuable suggestions. We have modified our paper accordingly. Please feel free to share your further feedback.

---

> ### Author Response · Authors · 2022-11-30
> **Responses to Reviewer 7vU5 part 2**
>
> Thanks again for reviewing our paper. Your main concern is about the motivation of LFT. We give empirical results and an explanation of the motivation behind the method in part 1. We are not sure whether we have addressed your concerns. We are very glad if we can assist with any further questions.
>
> You also have concerns about the novelty. We agree that the new methods proposed in this paper may share similar spirits with other works. However, we hope they are still fresh in the DETR fields. Moreover, all our new methods are well-motivated, as they are proposed based on observations of the drawback of previous methods. For example, contrastive denoising training adds negative samples to avoid duplicated predictions. Detailed ablations are conducted to verify the effectiveness of our methods. We hope the neat solutions can help to promote the development of DETR-like detectors.
>
> Thanks again. We are very glad if you can share your further feedback. If we have addressed your concerns, we hope you might consider updating your score.

---

### Official Review · Reviewer_eqZC · 2022-10-25

**Confidence:** 4
**Correctness:** 3
**Technical Novelty And Significance:** 3
**Empirical Novelty And Significance:** 3
**Recommendation:** 8

**Clarity, Quality, Novelty And Reproducibility:**

It is easy to follow the core ideas of the paper and the authors provided detailed explanation of experimental setup for effectively reproducing the results. Further, the authors promised to release their codebase. The paper has some simple and incrimental contributions. But those contributions are highly effective in achieving state-of-the-art performance.

**Details Of Ethics Concerns:**

The paper proposes a tranformer based end-to-end object detection framework. The paper has no major ethical concerns.

**Strength And Weaknesses:**

Strengths:

1.	Impressive results. Although previous DETR-like models were having end-to-end object detection pipelines, their results are inferior to classical CNN-based object detectors.  Unlike the previous DETR-like models, the paper reports state-of-the-art object detection results on the COCO benchmark.

2.	The model requires relatively less pre-training data and a smaller model size compared to many state-of-the-art methods in the benchmark.

3.	The proposed changes are simple, but effective in improving accuracy.

4.	The paper reports results on a large Swin-L backbone with large-scale pre-training.

5.	Thorough ablation experiments.


Weakness:

1.    Incremental novelty:  The key idea behind the proposed contributions such as contrastive denoising training, and look forward twice are well-known in the literature. I will not consider this as a major problem since the authors effectively adapted those ideas for improving performance on DETR framework.

2.   Please explain why the improvement on small objects is very high (+7.2% in Tab. 2) compared to medium and large-sized objects. Which of the proposed contribution helps in achieving this?  Why?

3.   Table 2 shows that the proposed method is faster (FPS 24) than existing DETR-based object detectors, and its speed is even comparable with deformable DETR. But it is not clear to me how the proposed model with many components can achieve a speed, comparable to deformable DETR, and faster than DN-DETR?

4.  In page 8 it is mentioned that “Note that the results of our models with ResNet-50 backbone are higher than those in the first version of our paper due to engineering techniques”. Please clarify which “first version” the authors are referring to”?

5. The paper utilizes many engineering tricks to achieve state-of-the-art results (Appendix F).

6.  In tables 2 and 4, what are the results of the proposed model without using additional engineering techniques mentioned in appendix F?

7. Table 4 shows that the proposed method achoves Impressive SOTA results. It uses smaller pre-training, it does not need mask annotation and requires less paramets. Still it will achieve results comparable with SOTA methods. For complete understanding of the model performance, it will be beneficial to introduce a comparison on  the inference time and FLOPS in Table 4.



**Summary Of The Paper:**

The paper proposes an end-to-end Transformer detector named DINO that archives impressive performance on the COCO benchmark. The proposed DINO model improves the training efficiency and the detection performance by utilizing (i) contrastive denoising training, (ii) look forward twice, and (iii) mixed query selection strategies. The experimental results show that  DINO outperforms all previous ResNet-50-based models on COCO val2017 in the 12-epoch and the 36-epoch settings using multi-scale features. Further, training DINO with a stronger backbone on a larger dataset leads to an impressive result of 63.3 AP on COCO 2017 test-dev which is comparable with even CNN-based state-of-the-art methods on mainstream detection task.

**Summary Of The Review:**

The paper has some incremental contributions. But those contributions are highly effective in achieving state-of-the-art performance on DETR framework. So, I am in favor of accepting this paper.

---

> ### Author Response · Authors · 2022-11-13
> **Responses to Reviewer eqZC part 1**
>
> Thanks for your encouraging comments on our model performance and experiments. We hope the proposed model can be served as a strong baseline for future research. We respond to your comments item by item in this section.
>
> ## 1.Concerns on our technique novelties.
> Thanks for your valuable comments. We agree that the new methods proposed in this paper may share similar spirits with other works. However, we hope they are still fresh in the DETR fields. Moreover, all our new methods are well-motivated, as they are proposed based on observations of the drawback of previous methods. For example, the contrastive denoising training adds negative samples to avoid duplicated predictions. Detailed ablations are conducted to verify the effectiveness of our methods. We hope the neat solutions can help to promote the development of DETR-like detectors.
>
> ## 2.About the large improvement on small objects.
> Thanks for your careful reading. There are several reasons for the large AP improvement on small objects (APs).
> - In Table 5, our optimized DN-Deformable DETR has APs of 28.2, which is +3.8 higher than that of the original DN-Deformable DETR in Table 2. The original one uses dense attention in the decoder, while the optimized one uses deformable attention, which is better for local attention and therefore improves APs. In addition, we fixed a problem in the original DN-Deformable DETR’s Transformer encoder---their deformable attention is not properly initialized using the initialization method in Deformable DETR. The Transformer encoder is a critical component that processes multi-scale image features, and multi-scale image features are critical for small objects. Therefore, the optimized one has higher APs.
> - In Table 5, we can see that CDN improves APs by 1.2. By introducing negative noised queries, CDN not only encourages the model to pick up the anchor nearest to the center of a GT box to make predictions but also explicitly suppresses farther anchors. Since small object detection is more sensitive to the quality of anchors, DINO with high-quality anchors can achieve better APs.
> - Query selection improves APs by 1.9. Query selection provides high-quality anchor initialization, which is especially beneficial to small objects. The reason is similar to reason 2 that small object detection is more sensitive to the quality of anchors.
>
> In summary, both engineering improvements and our proposed methods contribute to the large improvement of our model. Thanks again for your careful feedback. We have added the explanations to the revised paper.
>
> ## 3.Concerns about model speeds.
> Thanks for your question.  It is worth noting that all three techniques do not bring extra computation when inference compared with Deformable DETR. CDN is only used when training, which has no impact on inference. “Look forward twice” use the output anchor boxes before stopping gradients from computing loss, so it only affects training. Mix query selection uses learnable queries instead of selected queries. It also brings no computation cost compared with the query selection in Deformable DETR.
> As for the DN-DETR, DN-DETR uses dense attention (standard attention) while DINO adopts deformable attention, which is more efficient. Therefore, our proposed DINO is faster than DN-DETR while comparable with Deformable DETR.
>
> ## 4.The typo of "the first version" in our paper.
> Thanks for your careful reading and feedback. We are sorry that it is a typo in our paper. We have fixed it in our revised paper. Thanks again for providing the problem.
>
> ## 5. Concerns about engineering tricks in Appendix F.
> Thanks for your careful reading and valuable suggestions. We do agree that we have many improvements in model design and training, which help to reach a rather strong model. To demonstrate the work we did, we provide a detailed presentation of our improvements in Appendix F, hoping to share our explorations and results with the community.  We merge all engineering improvements into a strong baseline, named optimized DN-Deformable-DETR in row 1 of Table 5. All our new proposed methods are verified on the strong baseline. The results show that our three methods work well on such a  strong baseline, introducing +2.7 AP compared with the optimized DN-Deformable-DETR. We hope it is a fair comparison to demonstrate the effectiveness of our methods.

---

> ### Author Response · Authors · 2022-11-13
> **Responses to Reviewer eqZC part 2**
>
> ## 6.Concerns on model performance without engineer optimization.
> Thank you for your interesting question. These days, we did an experiment on the unoptimized model and found our method can also improve performance significantly. We show the results as follows.
> |                    | # epochs | AP   | AP$_{50}$ | AP$_{75}$ | AP$_S$ | AP$_M$ | AP$_L$ |
> |--------------------|----------|------|-----------|-----------|--------|--------|--------|
> | DN-Deformable-DETR | 12       | 43.4 | 61.9      | 47.2      | 24.8   | 46.8   | 59.4   |
> |      DINO raw      | 12       | 47.2 | 63.1      | 51.7      | 29.8   | 50.8   | 60.9   |
>
>
> ## 7.Concerns on the inference speed of the SOTA models:
> Thank you for your constructive questions. This is a common concern also raised by reviewer nKUz.
> These days, we conduct experiments to test the inference computation costs of our model, which are shown as follows.
> |Model| #Params(M) | GFLOPs | FPS |
> |:----------------------:|:------------:|:--------:|:-----:|
> |DINO-SwinL-5scale| 217.6 | 1284.5 | 8.1 |
> |DINO-SwinL-4scale| 217.2 | 703.5 | 12.8 |
>
> According to Table 1 in our paper, HTC(5-scale) with R50 has FPS as 5 which is even worse than our DINO-SwinL-5scale. Therefore, HTC with SwinG is certainly slower than DINO-SwinL-5scale. For other SOTA models such as Florence and BEiT-3, we cannot obtain the inference cost of their models on the leaderboard, since they are not publicly available. However, as these models have rather large numbers of parameters, e.g., SwinG 1900M v.s. DINO-SwinL-5scale 200M, we can speculate that their models are much more time-consuming than our DINO.
>
> Thanks again for your valuable questions. we have added the inference cost of our large models to the revised version.

---

### Official Review · Reviewer_nKUz · 2022-10-28

**Confidence:** 3
**Correctness:** 4
**Technical Novelty And Significance:** 4
**Empirical Novelty And Significance:** 4
**Recommendation:** 6

**Clarity, Quality, Novelty And Reproducibility:**

Overall I thought the quality of the article was good and clear. It has good originality and a significant solution to a hot topic.

**Strength And Weaknesses:**

Strength:
1. This paper investigates the DETR framework very well and gives a comprehensive survey of related works.
2. The performance is impressive. The advantages are mainly faster convergence, better performance and the ability to have smaller models.
3. The motivations of three different components are reasonable. And ablation experiments show the expected target has been achieved.

Weakness:
Although the model is only 1/10 of the swinT size, there is nothing result about the inference speed. So I wonder if the so-called small models don't actually have any advantage in terms of computational power.

**Summary Of The Paper:**

In this paper, the authors propose three different methods to improve the framework of DETR: a contrastive way for centre-surrounding, twice forward prediction for refining results and a mixed query selection. The experiments demonstrate the necessity of these methods and significant improvement in classical benchmarks compared with other methods.

**Summary Of The Review:**

Despite the lack of sufficient discussion on the speed of this article, I am still inclined to accept this paper.

---

> ### Author Response · Authors · 2022-11-13
> **Responses to Reviewer nKUz**
>
> Thanks for your encouraging comments on our motivations and performance. We hope our proposed DINO can help to promote the exploration of Transformer in detection fields.
>
> Your concern is about the efficiency of our model, which is an important point for applications. We compare the GFLOPS and FPS of basic models (with ResNet-50 backbones) in Table 2. The results show that our model outperforms all previous works under the same backbone and a similar efficiency.
>
> These days, we conducted some experiments to get the inference speed of our large model, DINO with SwinL. The results are shown below.
>
>
> |Model| #Params(M) | GFLOPs | FPS |
> |:----------------------:|:------------:|:--------:|:-----:|
> |DINO-SwinL-5scale| 217.6 | 1284.5 | 8.1 |
> |DINO-SwinL-4scale| 217.2 | 703.5 | 12.8 |
>
> Table R1.1. the computation cost of our models
>
> According to Table 1 in our paper, HTC(5-scale) with R50 has FPS as 5 which is even worse than our DINO-SwinL-5scale. Therefore, HTC with SwinG is certainly slower than DINO-SwinL-5scale. For other SOTA models such as Florence and BEiT-3, we cannot obtain the inference cost of their models on the leaderboard, since they are not publicly available. However, as these models have rather large numbers of parameters, e.g., SwinG 1900M v.s. DINO-SwinL-5scale 200M, we can speculate that their models are much more time-consuming than our DINO.
>
> Thanks again for your valuable questions. we will add the inference cost of our large models to the revised version.

---

> ### Author Response · Authors · 2022-11-30
> **Responses to Reviewer nKUz**
>
> Thanks again for reviewing our paper. Your concern is about our inference speed compared with other SOTA methods such as SwinV2-G-HTC++. Actually, our DINO-SwinL-5scale (our SOTA model) has FPS of 8.1 which is even higher than HTC-R50 according to Table 2 in our paper. SwinV2-G is far larger than Swin-L. Therefore, we are sure that our DINO-SwinL-5scale has higher inference speed than SwinV2-G-HTC++. We are not sure whether we have addressed your concerns. Please do let us know if we can assist with any further questions. If we have addressed your concerns, we hope you might consider updating your score.

---

### Author Response · Authors · 2022-11-13
**Summary of Paper Revision**

We appreciate all the reviewers for their valuable and constructive comments. We have revised our paper accordingly and summarized the main improvements below:
1. Add the inference speed and GFLOPS of large models to the Appendix. (by Reviewer nKUz)
2. Add explanations of AP improvements on small objects. (by Reviewer eqZC)
3. Fix the typos.  (by Reviewer eqZC, dnJ7 and 7vU5)
4. Add motivations of LFT in the Appendix.  (by Reviewer 7vU5)
5. Modify claims on model size in the abstract.  (by Reviewer 7vU5)
7. Add results of look forward three and four times in the Appendix. (by Reviewer 7vU5 and tCi7)
8. Add visualization of DINO and DN-DETR in the Appendix. (by Reviewer dnJ7)

---

### Decision · Program_Chairs · 2023-01-20

**Decision:**

Accept: poster

**Justification For Why Not Higher Score:**

In general, the paper got solid scores but reviewers listed a fair amount of concerns about the incremental novelty, e.g., 'the key idea behind the proposed contributions such as contrastive denoising training, and look forward twice are well-known in the literature' and concerns about lack of strong theoretical analysis and motivation regarding the look forward twice scheme. For these reasons, the paper is an interesting combination of existing techniques but the level of innovation and novelty could be greatly improved.

**Justification For Why Not Lower Score:**

Reviewers seem to agree that the proposed method is simple yet effective, the paper well organized, and the approach achieved strong experimental results over the state-of-the-art baselines.

**Metareview: Summary, Strengths And Weaknesses:**

The paper received high scores and maintained them after rebuttal which warrant a solid accept. Nonetheless, there were concerns about the incremental novelty, e.g., 'the key idea behind the proposed contributions such as contrastive denoising training, and look forward twice are well-known in the literature' and concerns about lack of strong theoretical analysis and motivation regarding the look forward twice scheme. On a less technical note, there exists DINO ViT model: perhaps it would make sense for authors to find a more unique to this work title. For these reasons, AC recommends to accept the paper as a poster.

**Note From Pc:**

if the above contains the word "oral" or "spotlight" please see: "oral" presentation means -> notable-top-5% and "spotlight" means -> notable-top-25%. As stated in our emails, we are disassociating presentation type from AC recommendations